# A Graph-Transformational Approach to Swarm Computation

**DOI:** 10.3390/e23040453

**Published:** 2021-04-12

**Authors:** Larbi Abdenebaoui, Hans-Jörg Kreowski, Sabine Kuske

**Affiliations:** 1OFFIS—Institute for Information Technology, Escherweg 2, 26122 Oldenburg, Germany; larbi.abdenebaoui@offis.de; 2Department of Computer Science, University of Bremen, P.O. Box 330440, D-28334 Bremen, Germany; kuske@uni-bremen.de

**Keywords:** swarm computation, graph transformation, cellular automata, particle swarms

## Abstract

In this paper, we propose a graph-transformational approach to swarm computation that is flexible enough to cover various existing notions of swarms and swarm computation, and it provides a mathematical basis for the analysis of swarms with respect to their correct behavior and efficiency. A graph transformational swarm consists of members of some kinds. They are modeled by graph transformation units providing rules and control conditions to specify the capability of members and kinds. The swarm members act on an environment—represented by a graph—by applying their rules in parallel. Moreover, a swarm has a cooperation condition to coordinate the simultaneous actions of the swarm members and two graph class expressions to specify the initial environments on one hand and to fix the goal on the other hand. Semantically, a swarm runs from an initial environment to one that fulfills the goal by a sequence of simultaneous actions of all its members. As main results, we show that cellular automata and particle swarms can be simulated by graph-transformational swarms. Moreover, we give an illustrative example of a simple ant colony the ants of which forage for food choosing their tracks randomly based on pheromone trails.

## 1. Introduction

The idea of swarm computation is to design systems that mimic the problem-solving behavior of swarms in nature like ant colonies, bee hives, bird flocks, fish schools, etc. One encounters quite a variety of swarm concepts and swarm algorithms in the literature (see, e.g., [1,2,3,4,5,6,7,8,9]). Moreover, there are several general computational approaches like cellular automata, particle swarms and ant colony optimization that are subsumed under the heading of swarm intelligence. In this paper, we propose graph-transformational swarms as a unifying framework using the methods of graph transformation. The notion of graph-transformational swarms is flexible enough to cover a variety of swarm concepts and provides a mathematical basis for the analysis of swarms with respect to correctness and efficiency. The hope is that different models of swarm computation can be better compared with each other within a common framework and that results for one model can be carried over to other models more easily. Moreover, the graph-transformational approach allows to employ graph-transformation tools for simulation, model checking and SAT solving in a standardized way.

A graph-transformational swarm consists of an arbitrary number of members of a finite number of different kinds. The members act simultaneously in a common environment which is represented as a graph. Moreover, there may be a cooperation condition to regulate the interaction and cooperation of the members as well as a goal to be reached. Kinds and members are modeled as graph transformation units (see, e.g., [10]) which are computational devices based on rules. The key is that the framework of graph transformation provides the concept of parallel rule application to formalize the simultaneous actions of swarm members. First ideas of graph-transformational swarms are presented by [11] and [12] where typical applications of ant colony optimization algorithms are modeled, but a general definition of graph transformational swarms is missing. A short draft version of this paper appeared as [13]. A good part of the paper is also integrated into the first author’s PhD thesis [14].

The paper is organized in the following way. In Section 2, the basic notions of graph transformation are recalled. Section 3 introduces graph-transformational swarms. In Section 4, an illustrating example is given: a simple ant colony the ants of which forage for food in a pheromone-driven manner. To demonstrate the power of our approach, we embed cellular automata in Section 5 and particle swarms in Section 6.

## 2. Graph Transformation

In this section, we recall the basic elements of graph transformation as far as needed in this paper (for more details, see, e.g., [15,16,17]). We consider directed edge-labeled graphs and their derivation by applications of rules. The graph transformation approach is chosen in such a way that rules can be applied in parallel and that their parallel applicability follows from the applicability of each of the involved rules and an additional independence condition. Moreover, we use the notion of graph transformation units which comprise a set of rules and a control condition. Such a unit is a computational device that models the derivation of graphs while the control condition is obeyed. Units are used as members of swarms, and the parallelism makes sure that the members can act simultaneously (cf. Section 3).

### 2.1. Directed Edge-Labeled Graphs

Let Σ be a set of labels with ∗∈Σ. A (*directed edge-labeled*) *graph* over Σ is a system G=(V,E,s,t,l) where *V* is a set of *nodes*, *E* is a set of *edges*, s,t:E→V and l:E→Σ are mappings assigning a *source*
s(e), a *target*
t(e) and a *label*
l(e) to every edge e∈E.

An edge *e* with s(e)=t(e) is a *loop*. If e∈E is labeled with *z*, *e* is also called a *z*-*edge* or a *z*-*loop* resp. An edge with label * represents an *unlabeled edge*. In drawings of graphs, the label * is omitted. The components *V*, *E*, *s*, *t*, and *l* of *G* are also denoted by $V_{G}$, $E_{G}$, $s_{G}$, $t_{G}$, and $l_{G}$, respectively. The empty graph is denoted by *∅*. The class of all directed edge-labeled graphs over Σ is denoted by $G_{\Sigma }$.

The *disjoint union* of two graphs *G* and *H* is defined as $G+H=(V_{G}⊎V_{H},E_{G}⊎E_{H},s,t,l)$ where ⊎ denotes the disjoint union of sets and for f∈{s,t,l}$f\left(e\right)=f_{G}\left(e\right)$ if $e\in E_{G}$ and $f\left(e\right)=f_{H}\left(e\right)$ otherwise.

For graphs $G,H\in G_{\Sigma }$, a *graph morphism*g:G→H is a pair of mappings $g_{V}:V_{G}\to V_{H}$ and $g_{E}:E_{G}\to E_{H}$ which are structure-preserving, i.e., $g_{V}\left(s_{G}\left(e\right)\right)=s_{H}\left(g_{E}\left(e\right)\right)$, $g_{V}\left(t_{G}\left(e\right)\right)=t_{H}\left(g_{E}\left(e\right)\right)$, and $l_{H}\left(g_{E}\left(e\right)\right)=l_{G}\left(e\right)$ for all $e\in E_{G}$. If the mappings $g_{V}$ and $g_{E}$ are inclusions, then *G* is called a *subgraph* of H, denoted by G⊆H. The *match* of *G* with respect to the morphism *g* is the subgraph g(G)⊆H.

### 2.2. Graph Transformation Rules

A *rule*
r=(L,K,R) consists of three graphs $L,K,\in G_{\Sigma }$ such that L⊇K⊆R. A *rule with positive context*
r=(C,L,K,R) consists of four graphs *C*, *L*, *K*, and *R* such that (L,K,R) is a rule and L⊆C. If *C* equals *L*, it is omitted in *r*. The components *C*, *L*, *K*, and *R* are called *positive context*, *left-hand side*, *gluing graph*, and *right-hand side*, respectively. Sample rules are always presented with the inclusion symbols so that left-hand side, gluing graph, right-hand side, and a possible positive context are clear from their positions. In order to avoid too much technical detail, we assume that the node sets of *L* and *K* are equal. This means that rule applications do not delete nodes. Figure 1 shows the rule *found* a variant of which is used in Section 4 for modeling a simple ant colony.

The gluing graph consists of two nodes, say *u* and *v*, as well as an unlabeled edge from *u* to *v* and a *food*-loop at *v*. The left-hand side consists of the gluing graph and an *A*-edge from *u* to *v*. The right-hand side consists of the gluing graph and an $A^{+}$-edge from *v* to *u* as well as an ϵ-edge from *u* to *v*.

Intuitively, the application of a rule (L,K,R) replaces an occurrence of *L* in some graph by *R* such that the occurrence of *K* is kept. Hence, the application of the rule *found* reverses an *A*-edge into an $A^{+}$-edge provided that it is attached to a node with a *food*-loop. Additionally, it inserts an ϵ-edge.

A rule with positive context (C,L,K,R) is applied in the same way as (L,K,R) provided that the occurence of *L* is located within an occurrence of *C*. If the left-hand-side of the rule *found* is regarded as positive context, we can remove the *food*-loop as well as the unlabeled edge from the remaining three rule components, because they are not changed. The result is displayed in Figure 2 where the two nodes of the gluing graph are numbered to fix their inclusion into the other graphs. It is worth noting that the rule in Figure 1 and the rule in Figure 2 are semantically equivalent.

Formally, the application of r=(L,K,R) with $V_{L}=V_{K}$ to a graph G=(V,E,s,t,l) consists of the following three steps.
(1)Choose a match g(L) of *L* in *G* subject to the *identification condition*, which requires that those items that are identified via *g* belong to the gluing graph *K*, i.e., $g_{E}\left(e\right)=g_{E}\left(e^{\prime }\right)$ for $e,e^{\prime }\in E_{L}$ implies $e=e^{\prime }$ or $e,e^{\prime }\in E_{K}$. (Without the identification condition, the Parallelization Theorem below would not hold.)(2)Remove the edges of $g_{E}\left(E_{L}\right)-g_{E}\left(E_{K}\right)$ and call the resulting graph *Z*.(3)Add the right-hand side *R* to *Z* by gluing *Z* with *R* in g(K) yielding the graph *H* with $V_{H}=V_{Z}⊎\left(V_{R}-V_{K}\right)$ and $E_{H}=E_{Z}⊎\left(E_{R}-E_{K}\right)$. The edges of *Z* keep their labels, sources, and targets so that Z⊆H. The edges of *R* keep their labels; they also keep their sources and targets provided that those belong to $V_{R}-V_{K}.$ Otherwise, they are redirected to the image of their original source or target, i.e., $s_{H}\left(e\right)=g\left(s_{R}\left(e\right)\right)$ for $e\in E_{R}-E_{K}$ with $s_{R}\left(e\right)\in V_{K},$ and $t_{H}\left(e\right)=g\left(t_{R}\left(e\right)\right)$ for $e\in E_{R}-E_{K}$ with $t_{R}\left(e\right)\in V_{K}.$

A rule with positive context r=(C,L,K,R) is applied to *G* in the same way provided that the morphism g:L→G can be extended to *C*. Figure 3 shows two applications of *found*.

An application of *r* to *G* w.r.t. the graph morphism *g* is denoted by $G\underset{r}{⟹}H$. It is called a *direct derivation* from *G* to *H*. The subscript *r* may be omitted if it is clear from the context. The sequential composition of direct derivations $G=G_{0}\underset{r_{1}}{⟹}G_{1}\underset{r_{2}}{⟹}\cdots \underset{r_{n}}{⟹}G_{n}=H$ (n∈N) is called a *derivation* from *G* to *H*. As usual, the derivation from *G* to *H* can also be denoted by $G\overset{n}{\underset{P}{⟹}}H$ where $\{r_{1},\cdots ,r_{n}\}\subseteq P$, or just by $G\overset{*}{\underset{P}{⟹}}H$. The string $\langle r_{1},\ldots ,r_{n}\rangle$ is the *application sequence* of the derivation. Figure 4 shows a derivation with application sequence 〈found,found〉.

Instead of applying found to the left upper *A*-edge and then to the right upper one, one can interchange the order which yields the same result with a different intermediate graph.

In the following, the class of all rules (with and without positive context) is denoted by R.

### 2.3. Parallel Rule Application

Let $r_{i}=\left(C_{i},L_{i},K_{i},R_{i}\right)\in R$ for i=1,…,n. Then the *parallel rule*
$p=\sum_{i=1}^{n}r_{i}=\left(\sum_{i=1}^{n}C_{i},\sum_{i=1}^{n}L_{i},\sum_{i=1}^{n}K_{i},\sum_{i=1}^{n}R_{i}\right)$ is given by the disjoint unions of the components. Figure 5 shows the parallel rule *found* + *found*. It can be applied to the left graph of Figure 4, if the *A*-edges are not identified (otherwise, the identification condition would be violated). The result is equal to the right graph of Figure 4 (see also Figure 6).

Let r=(C,L,K,R) and $r^{\prime }=\left(C^{\prime },L^{\prime },K^{\prime },R^{\prime }\right)$ be two rules and let $G\underset{r}{⟹}H$ and $G\underset{r^{\prime }}{⟹}H^{\prime }$ be two direct derivations w.r.t. the morphisms g:L→G and $g^{\prime }:L^{\prime }\to G$. Then the direct derivations are *parallel independent* if the corresponding matches intersect in gluing items only, i.e., $g_{V}\left(V_{L}\right)\cap g_{V}^{\prime }\left(V_{L^{\prime }}\right)\subseteq g_{V}\left(V_{K}\right)\cap g_{V}^{\prime }\left(V_{K^{\prime }}\right)$ and $g_{E}\left(E_{L}\right)\cap g_{E}^{\prime }\left(E_{L^{\prime }}\right)\subseteq g_{E}\left(E_{K}\right)\cap g_{E}^{\prime }\left(E_{K^{\prime }}\right)$.

The application of parallel rules and parallel independence are closely related as is shown by the well-known Paralllelization Theorem (see, e.g., [16,17] and Chapter 2 of [15]). This result is the basis of the simultaneous actions of members of graph-transformational swarms as introduced in the next section.

**Fact** **1**(Parallelization Theorem). For i=1,…,n, let $r_{i}=\left(C_{i},L_{i},K_{i},R_{i}\right)\in R$ and let $p=\left(C,L,K,R\right)=\sum_{i=1}^{n}r_{i}$ be the corresponding parallel rule. Then the following hold.
Let $G\underset{p}{⟹}X$ be a direct derivation w.r.t. g:L→G. Then there are direct derivations $G\underset{r_{i}}{⟹}H_{i}$ with the matching morphisms $g_{i}=g|L_{i}$ that are pairwise parallel independent where the morphism $g|L^{\prime }:L^{\prime }\to G$ denotes the restriction of *g* to $L^{\prime }$. for g:L→G and $L^{\prime }\subseteq L$.Let $G\underset{r_{i}}{⟹}H_{i}$ for i=1,…,n be direct derivations w.r.t. $g_{i}:L_{i}\to G$. Let each two of them be parallel independent. Then there is a direct derivation $G\underset{p}{⟹}X$ w.r.t. g:L→G defined by $g|L_{i}=g_{i}$ for i=1,…,n.

The theorem still holds for an infinite family $r_{i}\in R$ with i∈N.

According to the Parallelization Theorem, the rule components of *found*+*found* in Figure 6 can be applied separately to the left graph and are parallel independent. Conversely, these two parallel independent applications of *found* can be executed in parallel.

### 2.4. Control Conditions and Graph Class Expressions

Control conditions can reduce the nondeterminism of rule application. In more detail, each control condition *C* is defined over a finite set *P* of rules and specifies a set SEM(C) of derivations. The class of all control conditions is denoted by C. Control conditions can be composed by the operator & with $\mathrm{SEM}\left(C_{1}\;\&\;C_{2}\right)=\mathrm{SEM}\left(C_{1}\right)\cap \mathrm{SEM}\left(C_{2}\right)$ for all $C_{1},C_{2}\in C$.

A typical control condition is a priority relation > on a set *P* of rules meaning that a rule r∈P can only be applied if no other rule with higher priority is applicable. Another often used control condition is a regular expression over *P*. By definition, the constants empty, lambda and r∈P are regular expressions and the composites $e_{1};e_{2}$, $e_{1}|e_{2}$ and $e^{*}$ are regular expressions if $e_{1}$, $e_{2}$, *e* are regular expressions. A derivation obeys a regular expression *e* if the application sequence of the derivation belongs to the language of *e*. In other words, $e_{1};e_{2}$ allows a derivation if an initial section is allowed by $e_{1}$ and the remaining section by $e_{2}$; $e_{1}|e_{2}$ allows a derivation if $e_{1}$ or $e_{2}$ allows it; $e^{*}$ allows a derivation if it is a sequence of sub-derivations each allowed by *e*. The expression r∈P requires that *r* is applied; lambda allows any derivation of length 0; empty forbids any derivation. Alternatively to $r^{*}$, r! is used. It requests that *r* is applied as long as possible and not arbitrarily often.

All these examples of control conditions and their satisfaction apply not only to derivations over *P* but also to derivations of the form
$$G_{0}\underset{r_{1}+r_{1}^{\prime }}{⟹}G_{1}\underset{r_{2}+r_{2}^{\prime }}{⟹}\cdots \underset{r_{n}+r_{n}^{\prime }}{⟹}G_{n}$$
with $r_{1},\ldots .r_{n}\in P$ and $r_{1}^{\prime },\ldots ,r_{n}^{\prime }\in P^{\prime }$ if $\langle r_{1},\ldots ,r_{n}\rangle$ is an application sequence in the case of regular expressions or if, in the case of priorities for all i=1,…,n, $G_{i-1}\underset{r_{i}+r_{i}^{\prime }}{⟹}G_{i}$ implies $r_{i}\geq \hat{r}$ for all $\hat{r}\in P$ applicable to $G_{i-1}$.

Graph class expressions restrict the class $G_{\Sigma }$ to subclasses, i.e., each graph class expression *X* specifies a set $\mathrm{SEM}(X)\subseteq G_{\Sigma }$. The class of all graph class expressions is denoted by X. Typical examples of graph class expressions are graph properties like unlabeled with $\mathrm{SEM}\left(\mathrm{unlabeled}\right)=G_{\left\{*\right\}}$ or simple with $\mathrm{SEM}\left(\mathrm{simple}\right)=\{\left(V,E,pr_{1},pr_{2},pr_{3}\right)|E\subseteq V\times V\times \Sigma \}$ where $pr_{i}$ is the projection to the i-th component for i=1,2,3. Moreover, each graph $G\in G_{\Sigma }$ is a graph class expression with SEM(G)={G}. We also use required(X) and op(X) for X∈X as graph class expressions. SEM(required(X)) contains all graphs with a subgraph in SEM(X). SEM(op(X)) for some graph operator *op* contains all graphs obtained by the application of the operator to graphs in SEM(X). Explicit examples of such operators are *nest-looping* and *$food^{*}$-looping*. Applied to $G\in G_{\Sigma }$, the first operator adds one nest-loop to some node, and the second operator adds an arbitrary number of food-loops. Graph class expressions can be composed by the operator & with $\mathrm{SEM}\left(X_{1}\;\&\;X_{2}\right)=\mathrm{SEM}\left(X_{1}\right)\cap \mathrm{SEM}\left(X_{2}\right)$ for all $X_{1},X_{2}\in X$. Further graph class expressions are introduced where needed.

### 2.5. Graph Transformation Units

In the following we introduce a special case of graph transformation units, which is suitable for our purposes.

A *graph transformation unit* is a pair gtu=(P,C) where P⊆R is a set of rules, and C∈C is a control condition over *P*. The *semantics* of gtu consists of all derivations $G\overset{*}{\underset{P}{⟹}}H$ allowed by *C*.

A unit gtu is *related* to a unit ${\mathrm{gtu}}_{0}$ if gtu is obtained from ${\mathrm{gtu}}_{0}$ by relabeling. For a mapping rel:Σ→Σ, the relabeling of ${\mathrm{gtu}}_{0}$ is the unit $rel\left({\mathrm{gtu}}_{0}\right)=\left(rel\left(P_{0}\right),rel\left(C_{0}\right)\right)$ where the relabeling replaces each occurring x∈Σ in the components $P_{0}$ and $C_{0}$ of ${\mathrm{gtu}}_{0}$ by rel(x). The set of units related to ${\mathrm{gtu}}_{0}$ is denoted by $\mathrm{RU}({\mathrm{gtu}}_{0}).$

Each set P⊆R of rules induces a graph transformation unit specified by gtu(P)=(P,free) where free allows all derivations. For gtu({p}) with p∈R we write gtu(p) for short.

## 3. Graph-Transformational Swarms

In this section, we introduce graph-transformational swarms and their computations. The swarm members act simultaneously in a common environment represented by a graph. All the members of a swarm may be of the same kind or of different kinds to distinguish between different roles members may play. The number of members of each kind is given by the size of the kind. To increase the flexibility of this notion, we also allow multidimensional swarms by means of size vectors. In this case, the number of members of the respective kind is the product of the size components. Given a size vector $\left(n_{1},\ldots ,n_{l}\right)\in N_{>0}^{l}$, the index vectors $\left(i_{1},\ldots ,i_{l}\right)$ with $i_{j}\in \left[n_{j}\right]$ for j∈[l] are used to identify the members of the swarms, where $N_{>0}=N-\left\{0\right\}$ and [n]={1,…,n}. While a kind is specified as a graph transformation unit, the members of a kind are modeled as units related to the unit of this kind making sure in this way that all members of some kind are alike. A swarm computation starts with an initial environment and consists of iterated rule applications requiring massive parallelism meaning that each member of the swarm applies one of its rules in every step. In other words, each member acts sequentially according to its specification while all together are always busy. The choice of rules depends on their applicability and the control condition of the members. In some cases, a more restricted way of computation is reasonable. Hence, we allow to provide a swarm with an additional cooperation condition. Finally, a swarm may have a goal given by a graph class expression like the initial graphs are specified by such an expression. A computation is considered to be successful if an environment is reached that meets the goal.

**Definition** **1**(swarm). *A swarm is a system S=(I,K,s,m,c,g) where I is a graph class expression specifying the set of initial environments, K is a finite set of graph transformation units, called kinds, s associates a size vector $s\left(k\right)\in N_{>0}^{d(k)}$ with each kind k∈K where $d\left(k\right)\in N_{>0}$ denotes the dimension of the kind k, m associates a family of members ${\left(m{\left(k\right)}_{i}\right)}_{i\in [s(k)]}$ with each kind k∈K with $m{\left(k\right)}_{i}\in \mathrm{RU}\left(k\right)$ for all i∈[s(k)], c is a control condition called cooperation condition, and g is a graph class expression specifying the goal. For $s=\left(n_{1},\ldots ,n_{l}\right)\in N_{>0}^{l}$ and some l≥1, $\left[s\right]=\{\left(i_{1},\ldots ,i_{l}\right)|i_{j}\in \left[n_{j}\right],j\in \left[l\right]\}$.*

A swarm may be represented schematically as in Figure 7 where $s_{i}=s\left(k_{i}\right)$ and $m_{i}=m\left(k_{i}\right)$ for i∈[n].

**Definition** **2**(swarm computation). *A swarm computation is a derivation*
$$G_{0}\underset{p_{1}}{⟹}G_{1}\underset{p_{2}}{⟹}\cdots \underset{p_{q}}{⟹}G_{q}\;$$
*such that $G_{0}\in \mathrm{SEM}\left(I\right)$, $p_{j}=\sum_{k\in K}\sum_{i\in [s(k)]}r_{j_{ki}}$ with a rule $r_{j_{ki}}$ of $m{\left(k\right)}_{i}$ for each j∈[q], k∈K and i∈[s(k)], and c and the control conditions of all members are satisfied. For the satisfaction of the control condition of a unit, confer the definition for parallel derivations in Section 2.4.*


That all members must provide a rule to a computational step, is a strong requirement because graph transformation rules may not be applicable. In particular, if no rule of a swarm member is applicable to some environment, no further computational step would be possible and the inability of a single member stops the whole swarm. To avoid this global effect of a local situation, we assume that each member has the empty rule (∅,∅,∅) in addition to its other rules. The empty rule gets the lowest priority. In this way, each member can always act and is no longer able to terminate the computation of the swarm. In this context, the empty rule is called *sleeping rule*. It can always be applied, is always parallel independent with each other rule application, but does not produce any effect. Hence, there is no difference between the application of the empty rule and no application even within a parallel step.

To enhance the feasibility of the swarm concept, we allow also unbounded sizes, denoted by N or Z. In this case, we allow only computations where in each step all but a finite number of rules are empty. An example of a swarm with unbounded size is the swarm version of a cellular automaton in Section 5.

The concept of graph-transformational swarms provides a formal framework for the study of swarm computation. In many swarm approaches, the environments of the swarms are either chosen as graphs explicitly or can easily be represented by graphs. And because rules are widely and successfully used as the core of computation, graph transformation combining rules and graphs is a natural candidate for the formalization of swarm computation. The graph-transformational approach offers some advantages:Graphs and rules are mathematically well-understood and quite intuitive syntactic means to model algorithmic processes. Moreover, the additional use of control and cooperation conditions as well as graph-class expressions allows very flexible forms of regulation.Derivations as sequences of rule applications provide an operational semantics that is precise and reflects the computational intentions in a proper way.Based on the formally defined derivation steps and the lengths of derivations, the approach provides a proof-by-induction principle that allows one to prove properties of swarm computations like termination, correctness, efficiency, etc.In the area of graph transformation, one encounters several tools for the simulation, model checking and SAT-solving of graph transformation systems that can be adapted to graph-transformational swarms.And maybe most important, the Parallelization Theorem establishes a systematic and reliable handling of massive parallelism. In several swarm approaches, the simultaneous actions of swarm members are organized in a very simplistic way by avoiding any kind of conflict or are required, but not always guaranteed (cf. e.g., [18]). In contrast to that, the simultaneous actions of members of graph-transformational swarms is assured whenever the member rules are applicable and pairwise independent. Both can be checked locally and much more efficiently than the applicability of the corresponding parallel rule.

In the next three sections, we make an attempt to demonstrate the stated advantages by modeling three typical approaches to swarm computation.

## 4. A Simple Ant Colony

In this section, we illustrate the notion of graph-transformational swarms by modeling an ant colony the ants of which forage for food by mean of a simple pheromone mechanism. The sample graph-transformational swarm is presented in Figure 8.

The swarm consists of some ants all of the same kind. They act in directed graphs with a *nest*-loop and some food-loops. The node with the nest-loop has some further unlabeled loops that represent the actual food stock. All other initial edges are labeled by a positive integer representing a pheromone rate. We assume nest-food-connectedness meaning that the paths from the nest-looped node to some food-looped node visit all nodes. Moreover, we assume that the underlying environment graph is simple meaning that there are no parallel pheromone-labeled edges. This class of graphs is denoted by $\left(nest\;\&\;food^{*}\right)$-looping(simple&pheromone-labeled&nest-food−connected). During swarm computations further edges appear and disappear.

The kind ant defines the potential activities of an ant by means of five rules and some priorities. It can leave the nest by placing an *A*-edge and an ϵ-labeled edge in parallel to a pheromone-labeled edge with the nest-looped node as source. Then it can *forage* for food by walking through the graph passing one pheromone-labeled edge per step and placing a parallel ϵ-edge. The label *A* refers to the ant, and ϵ is an integer to be added to the pheromone value. If an ant reaches a food-node, then the rule found is applied changing the label *A* into $A^{+}$ and indicating in this way that the ant takes food. In this state, it moves back using the rule return until it can deliver which adds a food unit to the stock. Note that the returning ants pass edges from target to source so that the same paths are used as for foraging. Moreover, an ant leaves the amount ϵ of pheromone along the return paths too. The pheromone values of the passed edges are not updated immediately, but in the next computational step. This allows several ants to pass the same edge in the same step. The control condition requests some priorities. An ant can only leave the nest if it cannot do anything else, i.e., if neither the label *A* nor $A^{+}$ is around. In other words, it leaves the nest at the beginning and after each delivery. Moreover, foraging for food stops whenever food is found. And moving back stops whenever the nest is reached. Further control is provided by the labels $A^{+}$ and *A*. As long as *A* is present, only the rules forage and found may be applied. As long as $A^{+}$ is present, only return and deliver may be applicable. The application of found turns a foraging phase into a returning phase that ends with deliver.

Due to the nest-food-connectivity of the environmental graph, an ant can always act. If the *A*-edge points to a food-looped node, then rule found can and must be applied. Otherwise the *A*-edge has a target with another outgoing edge so that forage can be applied. If there is an $A^{+}$-loop, then return can be applied. To match the left-hand side of the rule in this case, its $A^{+}$-edge must be mapped to the $A^{+}$-loop. This is possible because matches are not assumed to be isomorphic images. If there is an $A^{+}$-edge pointing to the nest-looped node, then deliver can and must be applied. Otherwise, the $A^{+}$-edge points to a node with an incoming edge so that return can be applied. If all other fail, leave is allowed and possible.

The members of kind ant are obtained by relabeling *A* and $A^{+}$ by $A_{i}$ and $A_{i}^{+}$ resp. for i=1,…,n where *n* is the chosen size of the ant colony. All other labels are kept. As all rule applications remove only edges with labels $A_{i}$ and $A_{i}^{+}$, all rule applications are pairwise parallel independent if they concern different labels. In other words, the maximal parallel computation steps can be performed whenever an applicable rule is chosen for each ant. But there is one restriction given by the cooperation condition. It requires that ants act *pheromone-driven* meaning that the number of ants that pass an edge corresponds to the pheromone value of the edge. More precisely, let l be an ant that can pass the edges $e_{1},\ldots ,e_{k}$ with pheromone values $\phi_{1},\ldots ,\phi_{k}$ in the next step, then $e_{j}$ is used with the probability $\frac{\phi_{j}^{\alpha }}{\sum_{i=1}^{k}\phi_{i}^{\alpha }}$ where the parameter α can be chosen in a suitable way. The larger α is, the more the effect of the pheromone values is intensified in the heuristic choice.

The cooperation condition requires that after each action of the ants an update of the pheromone values takes place. The only member equals the kind and provides a single rule that adds ϵ to each pheromone-labeled edge for each parallel ϵ-labeled edge. The control condition requires that the update-rule is applied with maximal parallelism as long as possible. The applications of the update-rules are parallel independent if they update different pheromone-labeled edges. Therefore, update needs *m* steps where *m* is the maximum number of parallel ϵ-edges.

Finally, the goal specifies graphs where the stock, i.e., the number of extra loops at the nest-looped node, exceeds a given bound *b* that can be chosen freely.

From the description of this swarm, it is clear how the computations look like. The ants act in parallel each applying one of its five rules according to applicability and priority. In the first step, all ants leave the nest. Later in the computations, all five types of rules may occur simultaneously. After each ants action step, an update takes place. The alternation between ant action and update can go on for ever, but can be stopped if the stock is large enough. Will this event occur eventually? We assume that the initial graphs are nest-food-connected so that there are paths from the nest to each food-labeled node in particular. The ants use those paths with some probability depending on the pheromone values. Consequently, the ants come back to the nest after they found food with some probability so that the stock increases with some probability if the computation runs long enough and the number of ants is large enough. This can be guaranteed by assuming in addition that the initial environments are finite and cycle-free because then every ant finds food and returns to the nest eventually. The pheromone mechanism favors short paths before long ones. The fastest way to increase the stock is by running a shortest path from nest to food and back. Short paths get some extra pheromone earlier than long ones so that they will be used in the further computation with even higher probability. This reasoning shows that there is a correlation between the length of paths and the number of computation steps needed to fill the stock.

Because this is a very first example of graph-transformational swarms, we have kept it simple. In particular, the kind update could be designed in a more sophisticated way by adding evaporation rules. Moreover, the only member update could be replaced by update-members that are related to the pheromone-labeled edges so that the pheromone updating is also in the style of swarms.

We have implemented the *simple ant colony* swarm in the graph transformation tool GrGen.NET [19]. An experimental computation with a swarm of 20 ants is documented in Figure 9. For a better visualization, we omit the labels of the ants and replace the loops representing the food stock by a single loop labeled with the number of food units. The initial graph $H_{0}$ has 23 nodes including a node with a nest-loop and two nodes with food-loops. The initial pheromone values of all edges correspond to ϕ=1. In the probability function, we use α=2. The seven further displayed graphs $H_{i}$ for i∈{4,5,7,11,18,28,270} are the graphs after the *i*-th step of the ants and the following update each. The graph $H_{2}$ represents the resulting graph after four ant-steps. More precisely, in the first ant step all ants leave the nest, however the swarm is split in two groups from almost the same size 9 and 11. This is due to the pheromone-driven action of ants and the equal initial pheromone values. Afterwards all ants apply their forage-rules three times. The edges visited from each group can be easily recognized in $H_{4}$. Since their initial values are augmented by the underlying group’s number of members. The graph $H_{5}$ results after the 5th ant-step. One can see how all members go forward applying their forage rules again. However the group of 11 members splits in three subgroups when arriving in the node, say *u*, with three outgoing edges. In the 7th ant-step which generates $H_{7}$, a group of 5 ants find the food-node, say $f_{1}$, while all other ants forage further. In the 11th ant-step, 11 ants have found food and are returning to the nest. The other members still forage. $H_{18}$ displays the results of the 18th ant-step. The first ants have delivered 4 units of food, in addition one can see that the path between *u* and $f_{1}$ starts slowly to be preferred. In $H_{28}$ the ants have performed already 28 steps, and 20 units of food are delivered. The path between *u* and $f_{1}$ is frequently walked through meanwhile. $H_{270}$ displays the graph after 270 ant-steps with 337 food units. Based on the pheromone values, one can see that ants prefer the shortest path between the nest- and one of the food-nodes. The computation may be terminated whenever the chosen bound of the food stock is reached.

Our ant colony model is meant to exemplify how the features of graph-transformational swarms look like and work. How such models can be turned into applications that solve concrete optimization problems can be seen in [11,12].

## 5. Cellular Automata

Cellular automata are computational devices with massive parallelism known for many decades see, e.g., [20,21,22,23,24]. They are also considered as typical representatives of swarm computation [2]. In this section, we embed cellular automata into the framework of graph-transformational swarms.

A cellular automaton is a network of cells where each cell has got certain neighbor cells. A configuration is given by a mapping that associates a local state with each cell. A current configuration can change into a follow-up configuration by the simultaneous changes of all local states. The local transitions are specified by an underlying finite automaton where the local states of the neighbor cells are the inputs. If the network is infinite, one assumes a particular sleeping state that cannot change if all input states of neighbor cells are also sleeping. Consequently, all follow-up configurations have only a finite number of cells that are not sleeping if one starts with such a configuration.

To keep the technicalities simple, we consider 2-dimensional cellular automata the cells of which are the unit squares in the Euclidean plane



for all (i,j)∈Z×Z and can be identified by their left lower corner. The neighborhood is defined by a vector $N=\left(N_{1},\ldots ,N_{k}\right)\in {\left(Z\times Z\right)}^{k}$ where the neighbor cells of (i,j) are given by the translations $\left(i,j\right)+N_{1},\ldots ,\left(i,j\right)+N_{k}.$ If one chooses the local states as colors, a cell with a local state can be represented by filling the area of the cell with the corresponding color. Accordingly, the underlying finite automaton is specified by a finite set of colors, say *COLOR*, and its transition $d:\mathrm{COLOR}\times {\mathrm{COLOR}}^{k}\to \mathrm{COLOR}.$ Without loss of generality, we assume white∈COLOR and use it as sleeping state, i.e., $d(\mathrm{white},{\mathrm{white}}^{k})=\mathrm{white}.$ Under these assumptions, a configuration is a mapping S:Z×Z→COLOR and the follow-up configuration $S^{\prime }$ of *S* is defined by
$$S^{\prime }\left(\left(i,j\right)\right)=d\left(S\left(\left(i,j\right)\right),\left(S\left(\left(i,j\right)+N_{1}\right)\right),\ldots ,S\left(\left(i,j\right)+N_{k}\right)\right)).$$

If one starts with a configuration $S_{0}$ which has only a finite number of cells the colors of which are not *white*, then only these cells and those that have them as neighbors may change the colors. Therefore, the follow-up configuration has again only a finite number of cells with other colors than *white*. Consequently, the simultaneous change of colors of all cells can be computed. Moreover there is always a finite area of the Euclidean plane that contains all changing cells. In other words, a sequence of successive follow-up configurations can be depicted as a sequence of pictures by filling the cells with their colors.

**Example** **1.**
*The following instance of a cellular automaton may illustrate the concept. It is called SIER, has two colors, COLOR={white,black}, and the neighborhood vector is N=((−1,0),(0,1)) meaning that each cell has the cell to its left and the next upper cell as neighbors.*

*The transition of SIER changes white into black if exactly one neighbor is black, i.e., $d:COLOR\times COLOR^{2}\to COLOR$ with d(white,(black,white))=d(white,(white,black))=black and $d(c,\left(c_{1},c_{2}\right))=c$ otherwise.*

*If one starts with the configuration $S_{0}$ with $S_{0}\left(\left(10,0\right)\right)=S_{0}\left(\left(0,10\right)\right)=S_{0}\left(\left(30,0\right)\right)=S_{0}\left(\left(0,40\right)\right)=\mathrm{black}$ and $S_{0}\left(\left(i,j\right)\right)=\mathrm{white}$ otherwise, then one gets the configuration in Figure 10 after 50 steps.*

*Starting with a single black cell, SIER iterates the Sierpinski gadget (cf., e.g., [25]).*


Cellular automata can be considered as graph-transformational swarms. Let CA be a cellular automaton with the neighborhood vector
$$N=\left(N_{1},\ldots ,N_{k}\right)\in {\left(Z\times Z\right)}^{k},$$
the set of colors *COLOR* and the transition function $d:\mathrm{COLOR}\times {\mathrm{COLOR}}^{k}\to \mathrm{COLOR}.$ Then a configuration S:Z×Z→COLOR can be represented by a graph gr(N,S) with the cells as nodes, with an unlabeled edge from each cell to each of its neighbors and two loops at each cell where one loop is labeled with the color of the cell and the other loop with the coordinates of the cell. The set of all these graphs is denoted by G(CA).

If the color of a cell (i,j) changes, i.e., $d(S\left(\left(i,j\right)\right),\left(S\left(\left(i,j\right)+N_{1}\right),\ldots ,S\left(\left(i,j\right)+N_{k}\right)\right))\neq S\left(i,j\right),$ then the following rule with positive context

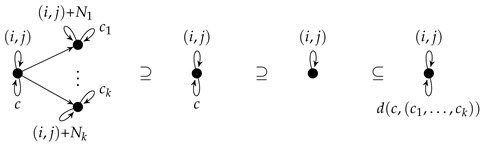

can be applied to the node (i,j) in gr(N,S) provided that c=S(i,j) and $c_{p}=S\left(\left(i,j\right)+N_{p}\right)$ for p=1,…,k. Due to the loops that identify the nodes, the matching is unique and the matches of the left-hand sides of each two of such applicable rules do not overlap. Consequently, all those applicable rules can be applied in parallel yielding $\mathrm{gr}(N,S^{\prime })$ where $S^{\prime }$ is the follow-up configuration of S. This remains true if the (empty) sleeping rule is applied to each other node because it is always applicable, is always independent of each other rule application and does not change the result. In other words, the derivation step $\mathrm{gr}\left(N,S\right)⟹\mathrm{gr}\left(N,S^{\prime }\right)$ is a swarm computation step if the rules above belong to members of a swarm which can be defined as follows:

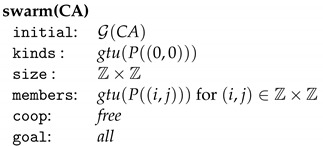

where the kind and the members are units induced by the sets of rules P((i,j)) containing all rules above for (i,j)∈Z×Z and the transition *d*. Every member gtu(P((i,j))) is obtained from the kind gtu(P((0,0))) by translating all points in the plane by (i,j) which is a special relabeling. Conversely, a computation step gr(N,S)⟹H in swarm(CA) changes a *c*-loop into a $d(c,\left(c_{1},\ldots ,c_{k}\right))$-loop at the node with the (i,j)-loop if and only if, for l=1,…,k, the neighbor with the $\left(i,j\right)+N_{l}$-loop has also a $c_{l}$-loop. All other c-loops are kept. This means that $H=gr(N,S^{\prime })$. Summarizing, each cellular automaton can be transformed into a graph-transformational swarm such that the following correctness result holds.

**Theorem** **1.**
*Let CA be a cellular automaton with neighborhood vector N and let swarm(CA) be the corresponding graph-transformational swarm. Then there is a transition from S to $S^{\prime }$ in CA if and only if $\mathrm{gr}\left(N,S\right))⟹\mathrm{gr}\left(N,S^{\prime }\right)$ in swarm(CA).*


Therefore, cellular automata behave exactly as their swarm versions up to the representation of configurations as graphs. We have considered cellular automata over the 2-dimensional space Z×Z. It is not difficult to see that all our constructions also work for the d-dimensional space $Z^{d}$ in a similar way. One may even replace the quadratic cells by triangular or hexagonal cells.

## 6. Particle Swarm Optimization

Particle swarm optimization is one of the major approaches to swarm intelligence one encounters in the literature in various variants (see, e.g., [26,27,28,29,30]) In this section, we model a discrete version of particle swarm optimization in the framework of graph-transformational swarms.

A particle swarm acts in the Euclidean space $R^{d}$ for some dimension d∈N. The space is provided with a fitness function $f:R^{d}\to R$ and a neighborhood $N:R^{d}\to P\left(R^{d}\right)$ (where P(X) denotes the power set of some set X). A swarm consists of *n* particles i∈[n] each of which carries the following information at each time t∈N: a *position*$p_{it}\in R^{d},$ a *velocity*$v_{it}\in R^{d},$ a *personal best (position)*
$pb_{it}\in R^{d},$ and a *best neighbor (position)*$bn_{it}\in R^{d}.$

The initial positions $p_{i0}$ and initial velocities $v_{i0}$ are chosen randomly. The initial personal bests coincide with the initial positions, i.e., $pb_{i0}=p_{i0}.$ In all steps, the best neighbor $bn_{it}$ is the position of a particle *j* in the neighborhood of $i,p_{jt}\in N\left(p_{it}\right),$ with maximum fitness, i.e., $f\left(p_{jt}\right)\geq f\left(p_{kt}\right)$ for all $p_{k}\in N\left(p_{it}\right).$ The positions, velocities and personal bests at time t+1 are given by the following formulas using the positions, velocities and personal bests at time *t*:$v_{i(t+1)}=v_{it}+U_{t}\left(0,\phi_{1}\right)⊗\left(pb_{it}-p_{it}\right)+U_{t}\left(0,\phi_{2}\right)⊗\left(bn_{it}-p_{it}\right),$$p_{i(t+1)}=p_{it}+v_{i(t+1)},$$pb_{i(t+1)}=p_{i(t+1)}$ if $f\left(p_{i(t+1)}\right)>f\left(pb_{it}\right)$ and $pb_{i(t+1)}=pb_{it}$ otherwise.

Here $\phi_{1}$ and $\phi_{2}$ are two pregiven bounds, $U_{t}\left(0,\phi_{1}\right)$ and $U_{t}\left(0,\phi_{1}\right)$ are vectors with randomly chosen components between 0 and $\phi_{1}$ and $\phi_{2}$ respectively and ⊗ is the componentwise product. A velocity represents a direction and a speed so that a particle moves in this direction with this speed from step to step where the velocity is adapted in such a way that the particle moves partly in the direction of the personal best and partly in the direction of the best neighbor. It is assumed that each particle is a neighbor of itself to guarantee that the best neighbor always exists. The goal is that one of the particles reaches a position the fitness of which meets or exceeds a given bound. In the literature, one can find a long list of examples of particle swarms which run successfully for a variety of optimization problems see, e.g., [28,29].

A simple way to discretize particle swarms is to assume that all position and velocity components and all randomly chosen scalars are integers. This discrete version of particle swarms can be transformed into the framework of graph-transformational swarms. Let PS be such a discrete particle swarm with the fitness function $f:Z^{d}\to Z,$ the neighborhood $N:Z^{d}\to P\left(Z^{d}\right),$ the bounds $\phi_{1},\phi_{2}\in N,$ the goal value b∈Z, and *n* particles. Then the corresponding graph-transformational swarm is given in Figure 11.

The initial environment graph is called space(N,f) and has all points $Z^{d}$ in the *d*-dimensional Euclidean plane with integer coordinates as nodes. There is an unlabeled edge (x,y) for $x,y\in Z^{d}$ with the source *x* and the target *y* whenever y∈N(x). Furthermore, each $x\in Z^{d}$ has two loops (x,1) and (x,2) where *x* is source and target. The label of (x,1) is also x, the label of (x,2) is f(x). All particles are of the same kind specified by the unit *particle* in Figure 12. (For technical simplicity, we assume d>1).

The member $particle_{i}$ for i∈[n] is obtained by indexing p,v,pb and bn with i. All other labels are kept variable with $p^{\prime },p^{\prime \prime }\in \left\{p_{1},\cdots ,p_{n}\right\}$ in particular. Due to the control condition, the rule *init* is applied first and then never again. It chooses two points *x* and *y*, generates a new node (representing a particle) and two edges from this node to *x* labeled with *p* and pb respectively and an edge to *y* labeled with *v* choosing randomly an initial position, which is also the personal best, and an initial velocity. As nothing is removed, each two applications of *init* are parallel independent such that all particles can be initialized simultaneously. Afterwards, a sequence of rule applications is iterated starting with *self* followed by *improve* as long as possible. The application of *self* takes the current position as best neighbor by adding a bn-edge parallel to the *p*-edge. The rule *improve* can be applied if one can find a particle in the neighborhood with a better fitness. Applied as long as possible, the bn-edge points to the current best neighbor.

If now *newvel* and then *newpos* are applied, then the velocity and position of a particle are changed using the formulas above by redirecting the *v*-edge and *p*-edge accordingly. If the new position has a better fitness than the former personal best, then the rule *newpb* can be applied to update the personal best. The control condition try(newpb) requires that newpb is applied if possible.

The rules *self* can be applied to all particles in parallel as again nothing is removed. Two applications of *improve* for different particles are parallel independent as only the different bn-edges are redirected. Therefore, the improvements can be done in parallel provided that at most one *improve*-rule per particle is applied. The cooperation condition requires that the applications of *newvel* are synchronized, which means that they are done in parallel after all improvements are performed. Each two applications of *newvel* for different particles are parallel independent as only different edges are redirected. Because of the same reason, all particles can get a new position by applying the *newpos*-rules in parallel. And analogously the *newpb*-rules can be applied in parallel afterwards as far as they are applicable at all. The cooperation condition requires that *self* is synchronized, which means that in the next round all applications of *self* start simultaneously. The goal requires that one of the particles reach a position the fitness of which meets or exceeds the bound value *b*.

The rules improve, newvel, newpos and newpb describe how the attributes of a particle can be changed by redirecting the respective edges where the positive context (placed left-most) provides the parameters that must be considered in each case.

By definition, a run ρ of a particle swarm is determined by the choices of $p_{i0}$ and $v_{i0}$ for i∈[n] and the vectors $U_{t}\left(0,\phi_{1}\right)$ and $U_{t}\left(0,\phi_{2}\right)$ for t∈N. The family of quadruples $s_{t}={\left(\left(p_{it},v_{it},pb_{it},bn_{it}\right)\right)}_{i\in [n]}$ may be seen as the swarm state at time t∈N. Such a state can be transformed into a graph $gr(s_{t})$ that has space (N,f) as subgraph and, for each i∈[n], an additional node *i* as well as four new edges of the form



Consider, on the other hand, the computation of swarm(PS) using the same choices as the run ρ. Then the considerations of this section show that, for each t∈N, the graph $gr(s_{t})$ is computed after all *improve*-steps in round *t* through the iteration in the control condition. This proves the following correctness result.

**Theorem** **2.**
*Let PS be a discrete particle swarm and swarm(PS) the corresponding graph- transformational swarm. Then there is a one-to-one correspondence between the runs in PS and the computations in swarm(PS).*


While particle swarm optimization is usually defined over a continuous space, we have transformed discrete versions of particle swarms into graph-transformational swarms because of the following reasons.
In the framework of graph transformation, the usual underlying structures are finite graphs or infinite discrete graph in exceptional cases. But all the concepts employed in the paper work for arbitrary sets of nodes and edges including the set of real numbers, the Euclidean space of some dimension or other continuous domains. Nevertheless, we have decided to consider a discrete version of particle swarm optimization as we want to demonstrate the potential of the usual graph transformation rather than to introduce a new kind of graph transformation. Nevertheless, the latter may be an interesting topic of future research.Moreover, implementations of particle swarm models are always discretized. As long as the abstract models are continuous, testing is the only way to validate an implementation against the model. A discrete abstract model between a continuous model and the implementation may allow to prove general properties and to improve the trustworthiness of system development in this way.In the literature, one encounters applications of particle swarm optimization to solve discrete problems (see, e.g., [30,31,32,33]). In such a case, a discrete abstract model seems to be appropriate. The particles correspond to problem solutions and the velocity and position updates, as introduced above, are redefined to be applicable to the discrete space. The graph-transformational model swarm(PS) above can also be adapted in the same way to solve discrete problems. In this case space(N,f) and the operators in the rule newvel should be adapted to the corresponding domains. Despite those changes all other components can be used unchanged

## 7. Conclusions

In this paper, we have introduced a graph-transformational approach to swarm computation providing formal methods for the modeling of swarms and the analysis of their correctness and efficiency. The concept exploits graph transformation units and the massive parallelism of rule applications.

As a first example, an ant colony with a simple pheromone-driven cooperation is modeled to illustrate the basic features of graph-transformational swarms. Our main results show that two other major approaches to swarm computation, cellular automata and particle swarms, can be embedded into the graph-transformational framework in a natural way.

The aim of this paper has been to advocate the syntactic and semantic concepts of graph-transformational swarms as a unifying framework for swarm modeling and analysis. To shed more light on the significance and usefulness of our approach, it would be of great interest to demonstrate that it does not only work on the abstract conceptual level, but also on the level of concrete real-world applications. To deliver convincing examples of this kind, quite some further work is needed and is a matter of future research. As first small steps in this direction, we refer to three papers where we consider potential applications concerning the solution of practical problems in cloud-based engineering systems [34] and in dynamic logistic networks with decentralized processing and control in [35] as well as of the routing problem of the automated guided vehicles in [36].

Future studies should provide further correct transformations from models with massive parallelism like ant colony optimization with more sophisticated pheromone-based computation, L-systems and DNA-computing into graph-transformational swarms. The hope is that graph-transformational swarms can serve as a common formal framework for a wide spectrum of swarm approaches.

## Figures and Tables

**Figure 1 entropy-23-00453-f001:**
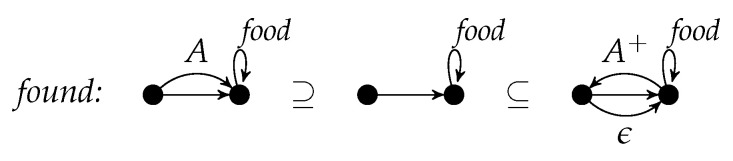
A graph transformation rule.

**Figure 2 entropy-23-00453-f002:**
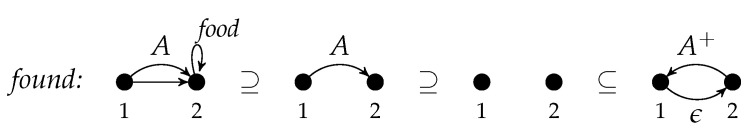
A graph transformation rule with positive context.

**Figure 3 entropy-23-00453-f003:**
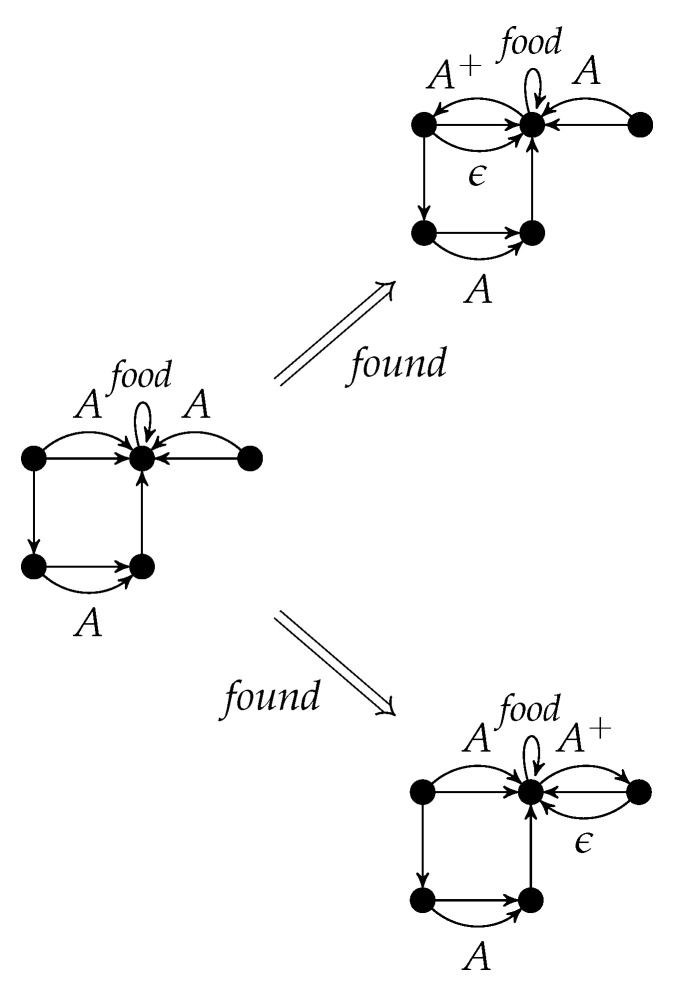
Two rule applications.

**Figure 4 entropy-23-00453-f004:**
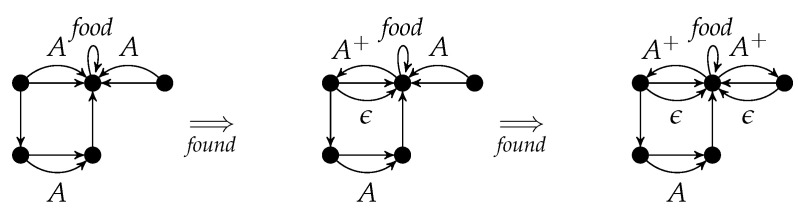
A derivation.

**Figure 5 entropy-23-00453-f005:**
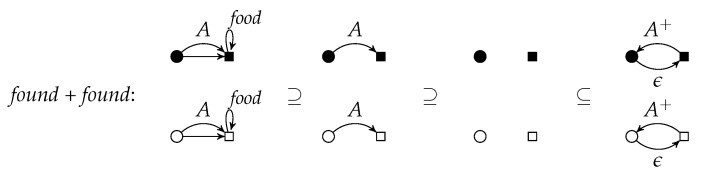
A parallel rule.

**Figure 6 entropy-23-00453-f006:**
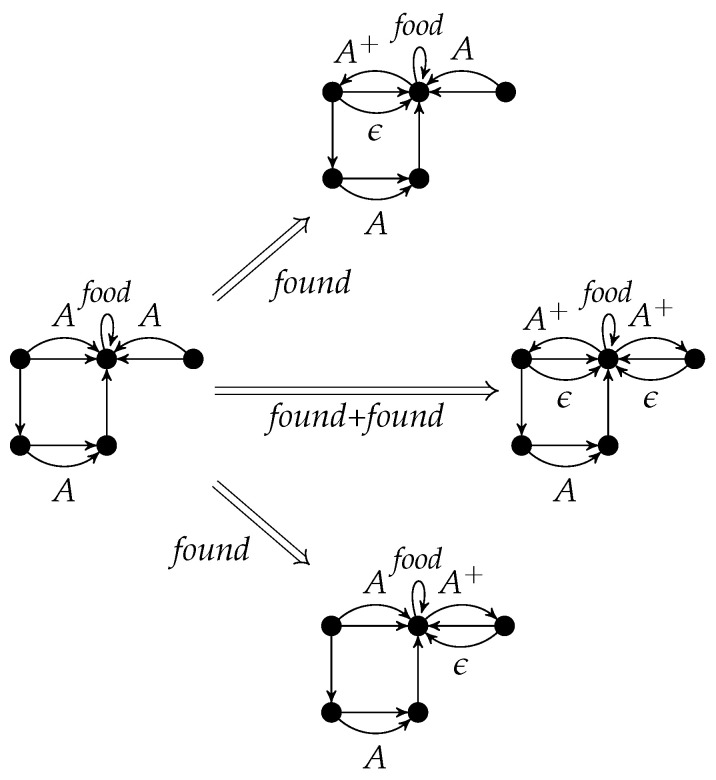
Two parallel independent rule applications and their parallelization.

**Figure 7 entropy-23-00453-f007:**
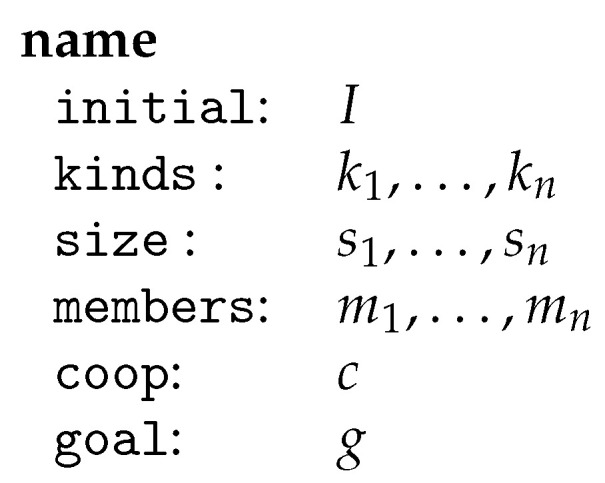
The schematic representation of a swarm.

**Figure 8 entropy-23-00453-f008:**
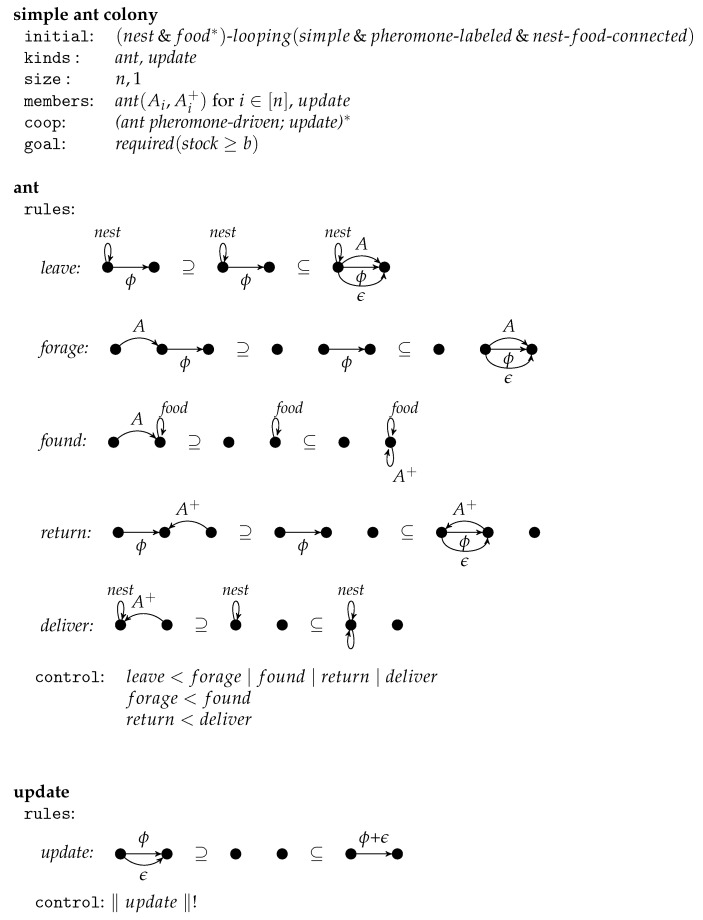
The swarm *simple ant colony* with the kinds *ant* and *update*.

**Figure 9 entropy-23-00453-f009:**
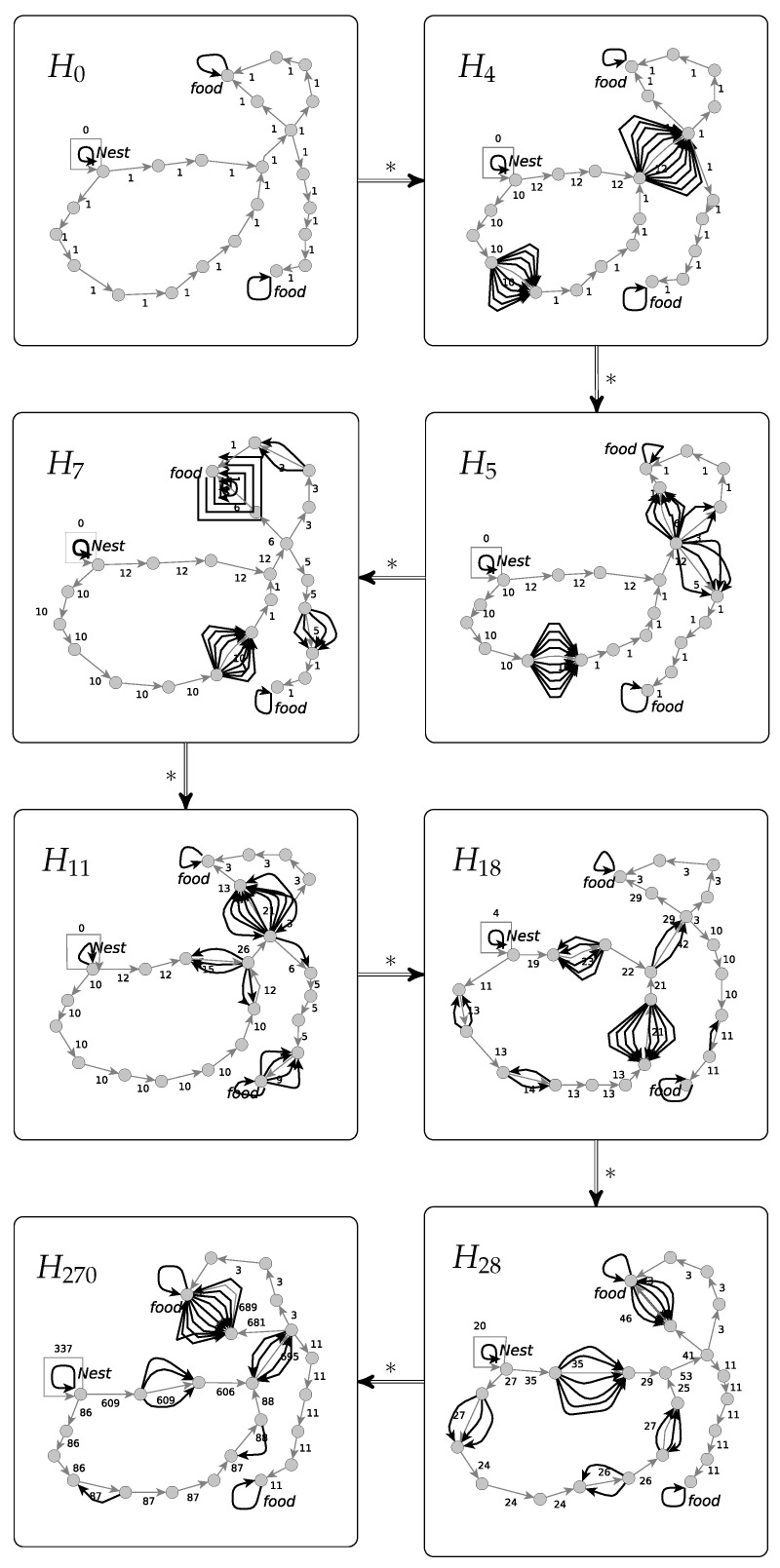
A sample computation of the *simple ant colony* swarm.

**Figure 10 entropy-23-00453-f010:**
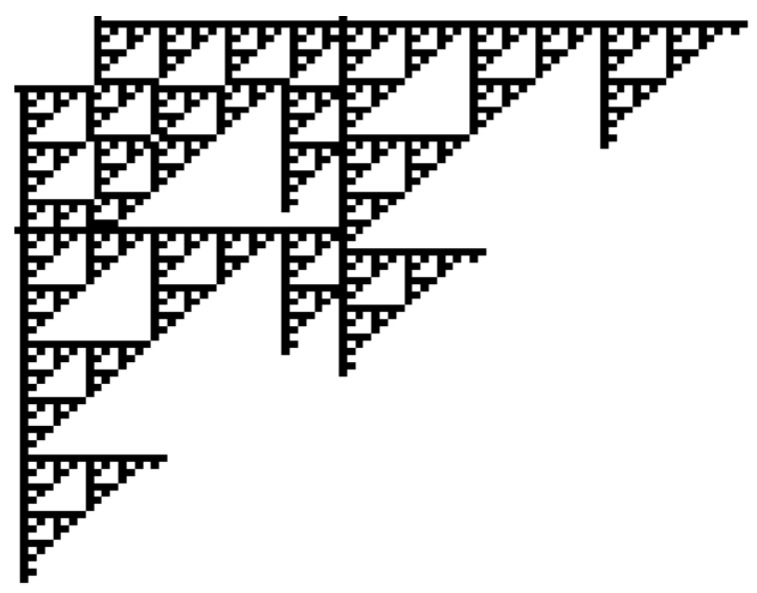
A pictorial representation of the configuration $S_{50}$.

**Figure 11 entropy-23-00453-f011:**
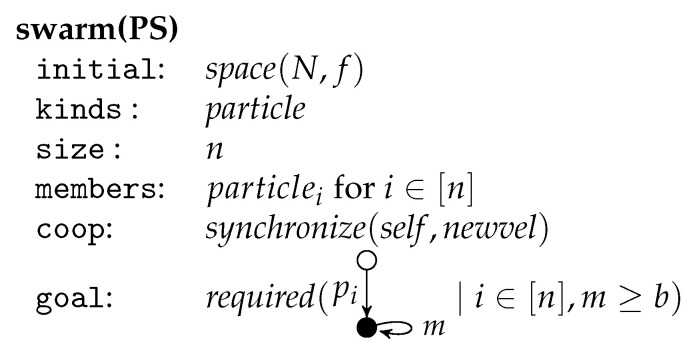
A graph transformational particle swarm.

**Figure 12 entropy-23-00453-f012:**
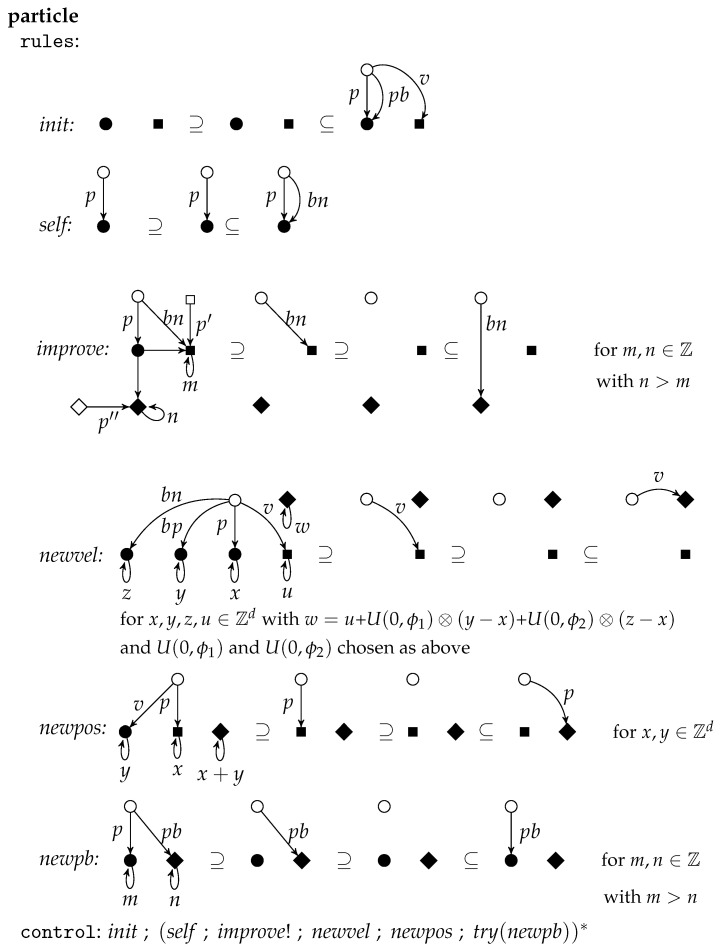
The unit *particle*.

## Data Availability

Data sharing not applicable.

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
