# Peer review of "A Graph-Transformational Approach to Swarm Computation"

_entropy, 2021, doi:10.3390/e23040453_

Round 1
Reviewer 1 Report
The manuscript presents a new formalism to represent and study swarm computation by means of graph transformations. That is, graph transformation rules encode the behavior of the swarms and the computation is performed by applying the rules. In this context, the environment in which the agents cooperate is represented by a graph.
The authors present three cases to prove their point: an ant colony (ACO), a cellular automata (CA) and particle swarm optimization (PSO).
The paper is well written, although there are some limitations that should be considered.
The authors provide a bottom up derivation of their conceptual framework, but unfortunately provide no proof for any of the theorems.
In the case of ACO and CA the idea of representing the computation as graph transformations seems sound and even logical. Unfortunately, this is not the the case of PSO. In particular:
- PSO is designed to work in real-valued spaces, so forcing a discretization of the candidate solutions (i.e., particle positions) is only useful to reduce the computation to a graph transformation. However, this is clearly not the case;
- PSO perturbs the velocity update using a random multiplier sampled with uniform distribution in [0,1]: again, the authors are changing the traditional algorithm to confirm their initial thesis;
- in conventional PSO, the best individual bn_it corresponds to the best individual appeared so far (i.e., up to the it-th iteration). On the contrary, in this manuscript, the authors propose a PSO in which the best individual is chosen from a neighborhood, which is a completely different definition (e.g., the particle could have been excellent in iteration it-1, and not-so-good in iteration it, so that it is no longer the global attractor). Again, this modification to the original algorithm seems to be motivated by the necessities of graph transformations.
I think that PSO does not belong to the class of swarm intelligence techniques that can be modeled using graph transformations. Thus, authors should clarify which algorithms and concepts can be implemented in this way.
Another concept that might be clarified is what happens if the rule transformation introduces the possibility of another rule to be fired. Stated otherwise, is it possible to have an infinite derivation?
Minors:
- Figure 1: please use labels to clarify what L, K and R are in the figure;
- Figure 2: the semantics of circles and squares are not clear. Why is the set of nodes bipartite? Please explain their roles, both in the text and caption;
- Figure 3 and 4: it is not clear why one rule is selected before the other. Is there a precedence/priority/ordering?
- Section 6: "(see, e.g., [24-26]) In this" < a point is missing.
Author Response
Dear Reviewer,
Thank you very much for your review. We have tried to consider your points of criticism as best as we can. In the following, we repeat each of the points and add our response in <<< ... >>>.
The authors provide a bottom up derivation of their conceptual framework, but unfortunately provide no proof for any of the theorems.
<<< Theorem 1 is a well-known fact In the area of graph transformation. Therefore, we address it now as “Fact” instead of “Theorem”. In Section 5, cellular automata and their transformation into graph-transformational swarms are formally introduced followed by a comparison and formal relation of their computational steps. Theorem 1 (formerly Theorem 2) states the result of this comparison. Similarly, in Section 6, particle swarms and their graph-transformational counterparts are introduced, their computational steps are compared, and the result is stated in Theorem 2 (formerly Theorem 3). We have presented the proving arguments before the theorems because we have wanted to stress the relation of the computational steps (assuming that many readers skip proofs).>>>
In the case of ACO and CA the idea of representing the computation as graph transformations seems sound and even logical. Unfortunately, this is not the case of PSO. In particular: PSO is designed to work in real-valued spaces, so forcing a discretization of the candidate solutions (i.e., particle positions) is only useful to reduce the computation to a graph transformation. However, this is clearly not the case;
<<< In the framework of graph transformation, the usual underlying structures are finite graphs or infinite discrete graphs in exceptional cases. But all the concepts employed in the paper work for arbitrary sets of nodes and edges including the set of real numbers, the Euclidean space of some dimension or other continuous domains. Nevertheless, we have decided to consider a discrete version of PSO as we want to demonstrate the potential of the usual graph transformation rather than to introduce a new kind of graph transformation (maybe an interesting future research topic for graph transformation). Moreover, implementations of PSO models are always discretized, As long as the abstract models are continuous, testing is the only way to validate an implementation against the model. A discrete abstract model between a continuous model and the implementation may allow to prove general properties and to improve the trustworthiness of system development in this way. Another aspect is that PSO is sometimes applied to discrete optimization problems even with finite particle domains. We have added a paragraph at the end of Section 6 to explain our choice of a discrete PSO model. >>>
PSO perturbs the velocity update using a random multiplier sampled with uniform distribution in [0,1]: again, the authors are changing the traditional algorithm to confirm their initial thesis;
<<< In the paper, our model is based on the common and general version of PSO that has been introduced as a metaheuristics in the book “swarm intelligence” by Kennedy et al. (reference [2]). Considering the special case, where the constants phi_1 and phi_2 are equal 1, we get a distribution in [0,1] as described in the basic version in Kennedy et al. 1995 (reference [26]). The underlying stochastic operations with the underlying perturbations are modeled in the rule newvel. >>>
in conventional PSO, the best individual bn_it corresponds to the best individual appeared so far (i.e., up to the it-th iteration). On the contrary, in this manuscript, the authors propose a PSO in which the best individual is chosen from a neighborhood, which is a completely different definition (e.g., the particle could have been excellent in iteration it-1, and not-so-good in iteration it, so that it is no longer the global attractor). Again, this modification to the original algorithm seems to be motivated by the necessities of graph transformations.
I think that PSO does not belong to the class of swarm intelligence techniques that can be modeled using graph transformations. Thus, authors should clarify which algorithms and concepts can be implemented in this way.
<<< For a given member in swarm(PS), the personal best position is used together with the best position in the neighborhood in order to update the current position. The personal best position is only updated if the current position is better than the former best personal position. Thus, the best position found so far in the swarm cannot get lost (see the control condition try(newpb)).Note that this version of PSO introduced in [2] is more general than the basic version proposed in [26] and it is introduced in order to avoid local minima. Actually, in the basic version there is no notion of neighborhood and the particles are attracted by the global best particle. If we consider in the general version the special case, where each particle has as neighbors all particles (also known as global topology) than we get exactly the basic version of PSO with a global attractor in the swarm. >>>
Another concept that might be clarified is what happens if the rule transformation introduces the possibility of another rule to be fired. Stated otherwise, is it possible to have an infinite derivation?
<<< We are not sure that we understand your remark properly. The application of rules yield usually results to which new rules can be applied. This is often desirable. Moreover, the derivation process may go on for ever (although we did not introduce this formally).
Minors:
Figure 1: please use labels to clarify what L, K and R are in the figure;
<<< We reserve the symbols C, L, K, R for the components of an arbitrary rule. But to make clearer which component is which in sample rules, the following sentence is added in line 80 after “respectively.”: Sample rules are always presented with the inclusion symbols so that left-hand side, gluing graph, right-hand side, and a possible positive context are clear from their positions. >>>
Figure 2: the semantics of circles and squares are not clear. Why is the set of nodes bipartite? Please explain their roles, both in the text and caption;
<<< The inclusion of the two nodes into left- and right-hand side of the rule must be fixed. Otherwise, the directions of the edges would not be clear. Therefore, the two nodes have different forms. To avoid confusion with bipartite graphs, we have replaced the different forms by a numbering of the two nodes. The sentence “where the different node shapes serve to identify the nodes of the gluing graph in the other components” is replaced by “where the two nodes of the gluing graph are numbered to fix their inclusion into the other graphs” >>>
Figure 3 and 4: it is not clear why one rule is selected before the other. Is there a precedence/priority/ordering?
<<< The application of a rule to a host graph is nondeterministic as there may be various occurrences of the left-hand side in a host graph. If there are more than one occurrence, then one has free choice. To demonstrate this, we show two possible rule applications: “Fig. 3 shows two possible applications of found” instead of “Fig. 3 shows an application of found”.>>>
Section 6: "(see, e.g., [24-26]) In this" < a point is missing.
<<< Done >>>
With kind regards,
Larbi Abdenebaoui, Hans-Jörg Kreowski and Sabine Kuske
Reviewer 2 Report
The paper proposes a very ingenious subject. The proposed approach is useful for a thorough understanding of swarm algorithms, facilitating the process for those with a strong formal background in formal languages. The computations that take place in the swarm are now formally presented and their parallelism is (more) obvious than in the traditional formulation of the algorithms. Hence, the paper would be of interest to those in need to have a unified view of the field of swarm algorithms.
It is impeccably written.
Author Response
Dear Reviewer,
Thank you very much for your review – quite encouraging.
With kind regards,
Larbi Abdenebaoui, Hans-Jörg Kreowski and Sabine Kuske
Reviewer 3 Report
The research adequately proposes a graph-transformational approach to a swarm, potentially improving its flexibility that is flexible and efficiency. The mathematical basis is appropriately presented.
However, the paper should be thoroughly revised to address the following comments:
- The authors are required to clearly explain how the proposed approach addresses the shortcomings and limitations of the swarm intelligence algorithms, e.g. of particle swarm optimization (PSO) algorithm. For a detail analysis of issues in applying PSO to address real life problems (e.g. process optimisation), see the paper: “Particle swarm optimisation in designing parameters of manufacturing processes: A review (2008–2018)”, Applied Soft Computing, https://doi.org/10.1016/j.asoc.2019.105743. Similar logic should be followed for the other major swarm intelligence algorithms, e.g. ACO.
- An illustrative example is given for the ant colony algorithm. However, the authors are asked to bring the whole approach closer to the practical implementation aspect. The authors should explained explain when / why / how the proposed research should be implemented in a real life / practice). What are the problem types that could be addressed by the proposed research? For a certain problem type, explain the difference of implementing the proposed approach and the conventional PSO (or ACO) approach? What are the benefits? What are the limitations of the proposed approach, particularly in terms of implementation?
- The literature review should be significantly extended and updated (the literature review covers the period from 1990-ties till 2010; the most recent paper is from 2013). The authors are requested to performed to address more recent studies (e.g. from the last 5 years). This in particular refers to the studies on the application of swarm intelligence algorithm (e.g. PSO) in a real life problems (e.g. industrial process optimization). The authors addressed the research on PSO application (paper “Analysis of the publications on the applications of particle swarm optimization”), but it is outdated – from 2008. Therefore, the literature review should be significantly updated in terms of the latest studies on the issues in implementing PSO to solve a real life problems. Based on that, the responses on the above comments 1 and 2 should be given.
Author Response
Dear Reviewer,
Thank you very much for your review. We have tried to consider your points of criticism as best as we can. In the following, we repeat each of the points and add our response in <<< ... >>>.
However, the paper should be thoroughly revised to address the following comments:
1. The authors are required to clearly explain how the proposed approach addresses the shortcomings and limitations of the swarm intelligence algorithms, e.g. of particle swarm optimization (PSO) algorithm. For a detail analysis of issues in applying PSO to address real life problems (e.g. process optimisation), see the paper: “Particle swarm optimisation in designing parameters of manufacturing processes: A review (2008–2018)”, Applied Soft Computing, https://doi.org/10.1016/j.asoc.2019.105743. Similar logic should be followed for the other major swarm intelligence algorithms, e.g. ACO.
<<< The framework of graph transformational swarms provides a mathematical basis for the analysis of swarms with respect to their correct behavior and efficiency. It brings also visualization advantages of the underlying computations. It is generally assumed, that in swarm inspired approaches, the members can act in parallel, however in some domains (such as smart grids, cyber physical systems or logistics) where errors are hardly or not at all allowed and where the members of swarms should have access to limited resources, the parallel behavior and computation of swarms play a critical role and their modeling and analysis is of high importance. To the best of our knowledge, the parallelism of swarms is only in few works considered explicitly see (Wrede et al. 2018)* and with high focus on the implementation. Our work focuses in general on the modeling aspects. Additionally, the visualization capabilities of our framework offer to designers and programmers to display the behavior on critical parts. One can have an overview of the developed solution and therefore easily analyze and debug but also to adapt and improve the solutions.
(Wrede et al. 2018) Wrede, Fabian, Breno Menezes, Luis F. Pessoa, Bernd Hellingrath, Fernando Buarque, and Herbert Kuchen. 2018. “High-level Parallel Implementation of Swarm Intelligence-based Optimization Algorithms with Algorithmic Skeletons.” >>>
2. An illustrative example is given for the ant colony algorithm. However, the authors are asked to bring the whole approach closer to the practical implementation aspect. The authors should explained explain when / why / how the proposed research should be implemented in a real life / practice). What are the problem types that could be addressed by the proposed research? For a certain problem type, explain the difference of implementing the proposed approach and the conventional PSO (or ACO) approach? What are the benefits? What are the limitations of the proposed approach, particularly in terms of implementation?
<<< The aim of this paper has been to advocate the syntactic and semantic concepts of graph-transformational swarms as a unifying framework for swarm modeling and analysis. Our emphasis has been laid on the embedding of three well-known approaches to swarm modeling into our framework. Although it would be of great interest to demonstrate that this does not only work on the abstract conceptual level, but also on the level of concrete real-world applications, we are not able to deliver convincing examples without quite some further research. As our expertise is mainly on formal modeling methods and theoretical computer science, we are not able to provide new results in this respect in a few days. As a small step in this direction, we discuss the matter in the conclusion as future work. Moreover, we refer to three papers where we consider potential applications (solving practical problems in cloud-based engineering systems [34] and in dynamic logistic networks with decentralized processing and control in [35] as well as proposing a solution to the routing problem of the automated guided vehicles in [36]). Moreover, we have added small paragraphs at the ends of Section 4 and Section 6 to indicate that our considerations concern the abstract level of modeling rather than application and implementations. >>>
3. The literature review should be significantly extended and updated (the literature review covers the period from 1990-ties till 2010; the most recent paper is from 2013). The authors are requested to performed to address more recent studies (e.g. from the last 5 years). This in particular refers to the studies on the application of swarm intelligence algorithm (e.g. PSO) in a real life problems (e.g. industrial process optimization). The authors addressed the research on PSO application (paper “Analysis of the publications on the applications of particle swarm optimization”), but it is outdated – from 2008. Therefore, the literature review should be significantly updated in terms of the latest studies on the issues in implementing PSO to solve a real life problems. Based on that, the responses on the above comments 1 and 2 should be given.
<<< We have updated the list of references by adding the references 6, 7, 8. 9. 14. 29, 30, 34, 35, and 36. >>>
With kind regards,
Larbi Abdenebaoui, Hans-Jörg Kreowski and Sabine Kuske
Round 2
Reviewer 3 Report
The authors appropriately revised the paper according to the reviewer comments.